



# The origin of hydrological responses following earthquakes in confined aquifer: Insight from water level, flow rate and temperature observations

Shouchuan Zhang[1,2], Zheming Shi[1*], Guangcai Wang[1], Zuochen Zhang[3], Huaming Guo[1]

[1]MOE Key Laboratory of Groundwater Circulation and Environmental Evolution, China University of Geosciences, Beijing, China
[2] Chinese Academy of Geological Science, Beijing, China
[3]Institute of Geomechanics, Chinese Academy of Geological Science, Beijing, China

*Corresponding to:* Zheming Shi (szm@cugb.edu.cn)

**Abstract.** Although many mechanisms of earthquake-induced hydrological response have been proposed in recent decades, the origins of these responses remain enigmatic and a quantitative understanding of them is lacking. In this study, we quantitatively analyzed water level, flow rate, and water temperature data from an artesian well in southwestern China before and after multiple earthquakes. Water level and temperature always showed co-seismic step-like increases following earthquakes, which were independent of the

earthquakes' epicentral distances and magnitudes. Tidal analysis found changes in aquifer and aquitard permeability following these earthquakes, which correspond to the post-seismic discharge of 85~273 m$^3$. Furthermore, we coupled the flow rate and temperature data to model the mixing processes that occurred following each earthquake. The results indicate that co-seismic temperature changes are the result of the mixing of different volumes of water from shallow and deep aquifers, with the mixing ratio varying according

to each earthquake.

## 1 Introduction

Earthquake-induced hydrological anomalous phenomena, such as changes in water levels (Hosono et al., 2019; Weaver et al., 2019; Zhang et al., 2019a; Zhang et al., 2021), spring discharge (Manga and Rowland, 2009; Manga et al., 2016; Yan et al., 2020; Mohr et al., 2017), water temperature (Miyakoshi et al., 2020;

Chien et al., 2020; Wang et al., 2012) and chemical composition (Shi et al., 2020; Skelton et al., 2014a; Kim et al., 2019; Igarashi et al., 1995; Skelton et al., 2019; Li et al., 2019; Zhang et al., 2020) have been widely documented. Mechanisms proposed to explain these phenomena include (1) water expulsion by co-seismic static strain, (2) water release from shallow crustal areas due to increased permeability following earthquakes, and (3) increased water recharge from a storage source due to an increased hydraulic head gradient. However,

most of these mechanisms have been proposed depending either on observations of co-seismic streamflow or water level responses, which makes it difficult to identify the origins of the release of co-seismic groundwater. These mechanisms usually do not quantitatively explain the magnitude of co-seismic responses. Furthermore, the absence of integrated multi-parameter explanations for the proposed mechanisms may lead to inconsistent or even contradictory explanations of different co-seismic hydrological phenomena. Although



several studies have documented earthquake-induced changes in groundwater level, flow rate, and temperature, quantitative couplings of earthquake-induced groundwater flow and temperature are lacking, resulting in ambiguity in identification of the origins of earthquake-induced hydrological responses (Itsuo et al., 1988; Anderson, 2010; Wang and Manga, 2010).

In this study, we observed water levels and flow rate responses in an artesian well (Dazhai well) to quantitatively evaluate the mechanism of anomalous hydrological changes. Water temperature monitoring data were used to constrain the origin of the earthquake-induced hydrological response.

## 2 Geological Setting and Data Sources

Dazhai well (DZ well) is located in the Yixiangba sub-basin, 11 km southeast of Pu'er City in Western Yunnan Province, Southwest China. The Yixiangba sub-basin belongs to the Lanping-Simao basin, which is a rift basin situated in the southern section of the Sanjiang fold system. It is located in the southern Yunnan-Tibet geothermal zone, one of the most active areas of tectonic activity in China. The Lanping-Simao basin is underlain by strata ranging in age from Proterozoic to Cenozoic. The outcropping strata are related to the division of tectonic units of the basin. Due to the later denudation, the Upper and Middle Paleozoic limestones outcrop in the central and marginal uplifts of the basin. The Proterozoic metamorphic rocks occur along the Jinshajiang-Ailaoshan fault in the east part of the basin.

Tectonically, the Lancangjiang fault and Jinshajiang-Ailaoshan Deep faults are the western and eastern boundaries of the Lanping-Simao basin (Deng et al., 2020). The Wuliangshan Fault developed in the central basin, which is composed of several secondary faults, including Mohei, Pu'er East, Pu'er West, Zhenyuan-Puwen East, and Zhenyuan-Puwen West faults. These secondary faults are divided into several segments by a series of NE-trending faults. Faults zone can modulate fluid circulation within the shallow crust acting as conduits for groundwater flow. Veins and microfractures along faults caused by the crust dilation and contraction allows fluid re-distribution and mixing from different reservoirs (Smeraglia et al., 2018). A series of faults control the tectonic evolution of the rifting basin and also provide favorable conditions for groundwater storage and flow (Zhang et al., 2019b). As is shown in Figure 1, the aquifers are classified into five categories, including Loose rocks aquifer, Intrusive rock aquifer, Metamorphic rock, Clastic rock aquifer and Carbonate rock aquifer. Most of the aquifer type in the basin belongs to Clastic rock aquifer. The Loose rocks and Carbonate rock aquifer occur sporadically in the northern and southern part of the basin respectively. The Metamorphic rock and Intrusive rock aquifer are widely distributed in the east part of the basin.

DZ well (constructed in 1984 and upgraded to a digitized observation system in 2001) is located in fractured sandstone-siltstone rock in the Jinmenkou-Shigaojing part of Pu'er West fault (Figure 1). DZ well is an artesian well with a discharge rate of 0.3–0.6 L/s, a depth of 112.27 m with a casing diameter of 150 mm, a blank casing from 0–26.69 m with an open section between 26.69–112.27 m, and diameters of 120 mm and 110 mm at depths of 26.69–83.7 m and 83.7–112.27 m, respectively. Groundwater level, flow rate and water temperature are observed in DZ well. The lithology of the wellbore in DZ well is as follow (Figure



2): 0~7.51m is quaternary deposit, 7.51m~14.03m is argillaceous siltstone with strong weathering, 14.03m~26.69m is silty mudstone, 26.69~78.90m is argillaceous siltstone, 78.90m~110.76m is detrital sandstone with fissure developed, and 110.76m~112.27m is mudstone. In order to understand the deeper lithology and the distribution of aquifers, we also collected the aquifer lithology and wellbore structure of

Dazhai deep well (Figure S1 in the supporting information), which is drilled with a depth of 410 m and only 50m away from Dazhai well. Both wells are artesian wells. Sandstone has the characteristic of high mechanical strength and permeability, which is considered as the important aquifer in this area. On the contrary, the characteristic of mudstone is low mechanical strength and permeability, which is considered as aquitard. In combining these two wellbores, we could find that sandstone stratum is separated by the

mudstone at depth between 105-117m, formed two confined aquifers, and the groundwater head in the underlying sandstone aquifer (1480m) is higher than the shallow sandstone aquifer (1475m).

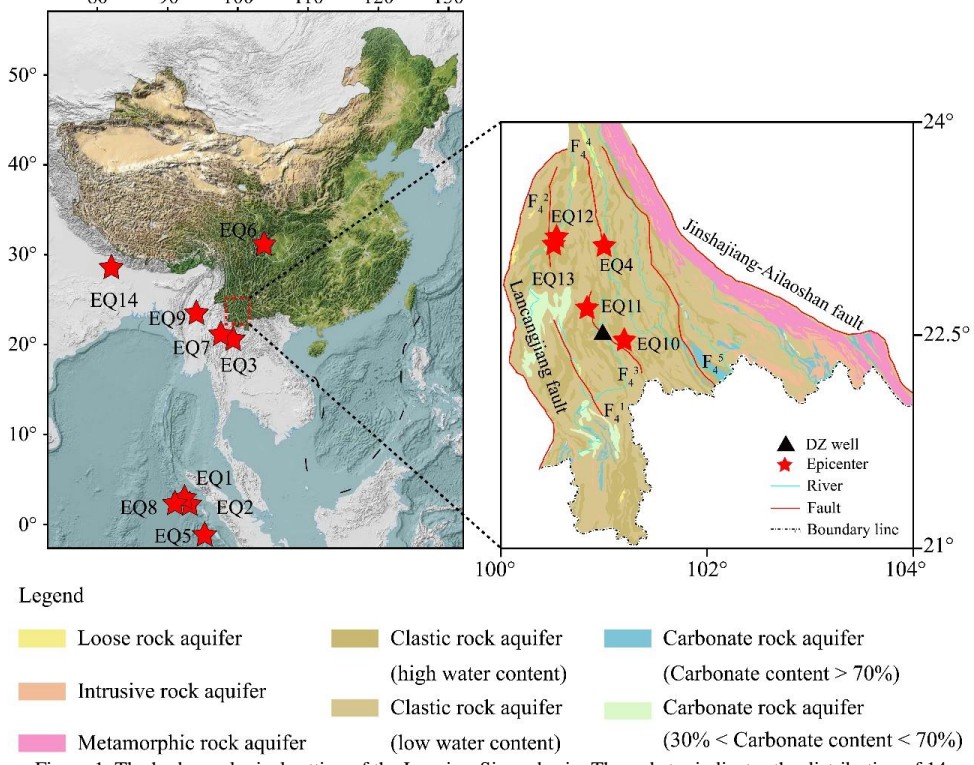

Figure 1. The hydrogeological setting of the Lanping-Simao basin. The red star indicates the distribution of 14 earthquakes, the black triangle indicates the DZ well, and the red solid lines indicate faults.

F4: The Wuliangshan fault, $F_4^1$: The Zhenyuan-puwen west fault, $F_4^2$: The Zhenyuan-puwen east fault, $F_4^3$: The Pu'er west fault, $F_4^4$: The Pu'er east fault, $F_4^5$: The Mohei fault.

The earthquake groundwater monitoring wells in China are categorized as two types: (1) static water level observation for the non-artesian well and (2) dynamic water-level observation for the artesian well (Cea,

2012). The DZ well is an artesian aquifer and the hydraulic head is always above the surface. Thus, the dynamic water level observation is employed. Such groundwater monitoring system has been widely used in





Chinese earthquake groundwater monitoring well in artesian aquifer, and they showed more sensitive response to > *Mw* 5.0 earthquakes than the static water level observation (Wan et al., 1990; Che, 1990). In order to observe the variation of water-level fluctuation, the wellbore is constructed above the ground and

equipped with the discharge port (Figure 2). The purpose of setting discharge port is to discharge a part of groundwater and keep the water level fluctuate in a certain range. Meanwhile, the discharge can be measured through this port by using the stopwatch capacity method, i.e., the required time per unit volume of water is measured. Without the discharge port, the hydraulic head would be 4 meters above the surface, which is difficult for the monitoring. Piezometric tube is used to calibrate the data of water level. For such a monitoring

system, the water level refers to the distance between the water table and the center line of the discharge port. Dynamic water level observation in DZ well is recorded hourly with a resolution of 1 mm using a LN-3A pressure transducer (LN-3A, developed by the Institute of Earthquake Forecasting, China Earthquake Administration, Beijing, China). Water temperatures are digitally recorded every minute using a quartz crystal thermometer (SZW-1A, developed by the Institute of Crustal Dynamics, China Earthquake

Administration, Beijing, China) with a resolution of 0.1 mK (Shi and Wang, 2014). The temperature probe is set at a depth of 100 m, removed from the influences of other factors such as air temperature, precipitation, and pumping. Monitoring data are transferred to and stored at the Yunnan provincial earthquake administration, Kunming, China.

We collected water level, flow rate and water temperature data for the period 2004–2015. Water levels

show seasonal fluctuations with a higher level during the summer wet season and lower levels during the drier winter season. The result of stable-isotope indicate recharge is primarily from meteoric waters (Yang et al., 2005).

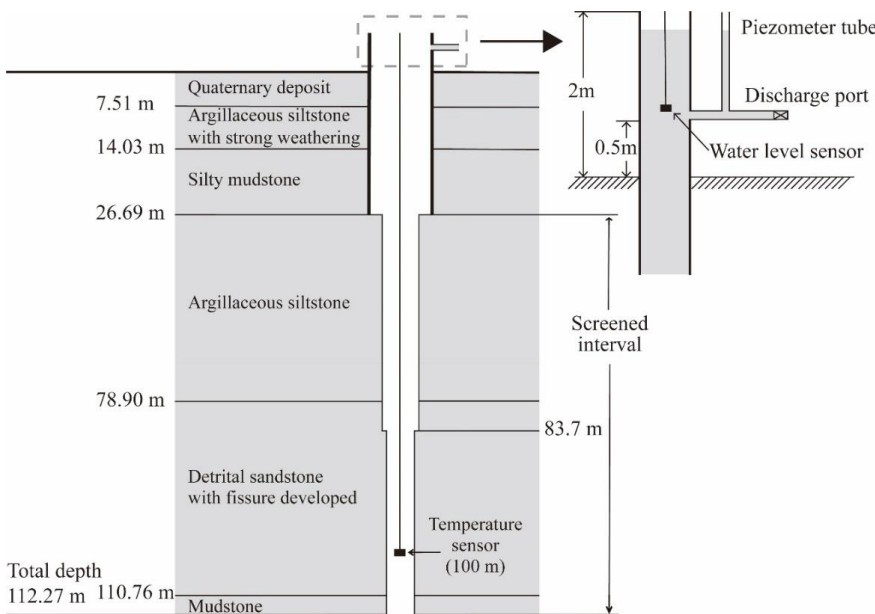

Figure 2. Aquifer lithology and specifications of DZ well aquifer system.





## 3 Coseismic Water-level and Temperature Changes

Three patterns of water level coseismic response have been identified based on the distance between the length of the ruptured fault and the epicentral distance: (1) Step-like changes in the near-field (< 1 ruptured fault length), (2) gradual changes, both increases and decreases, at intermediate distance (1–10 ruptured fault length), and (3) oscillations at far-field (> 10 ruptured fault length). During the 2004 to 2015 monitoring intervals, these responses were observed for fourteen earthquakes, including the 2004 *Mw* 9.0 Sumatra earthquake, 2008 *Mw* 7.9 Wenchuan earthquake, and 2015 *Mw* 8.2 Nepal earthquake (Figure 1). There is no barometric pressure monitoring data of 2004~2015 in the monitoring site of DZ well thus no barometric pressure correction had been done for the water level. The raw data of water level is shown in Figure S2 of the supporting information. We have removed the effect of earth tide in calculating the co-seismic water level changes. The co-seismic and post-seismic changes in water level and temperature can be clearly identified (Figure 3), and show sharp step-like rises, with anomalous changes following earthquakes lasting approximately a month (summarized in Table 1).

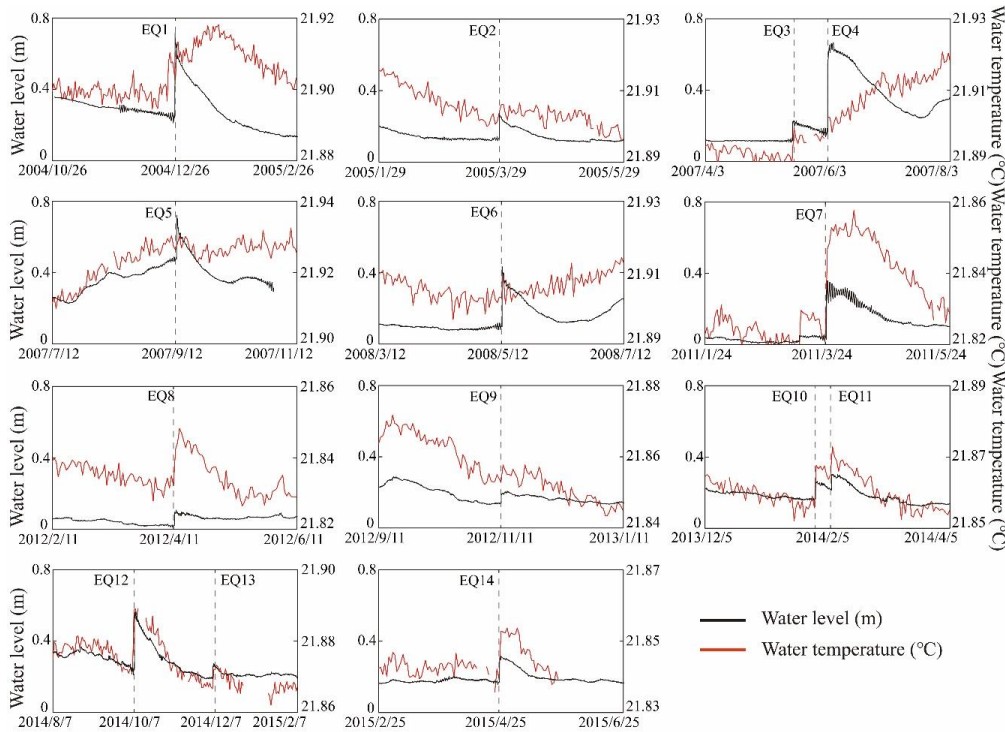

Figure 3. Water level and temperature changes induced by fourteen earthquakes in Dazhai well, western Yunnan Province, China. The water level changes have been corrected for Earth tide. Vertical dashed lines indicate earthquake times.

All water level coseismic responses show step-like increases, with magnitudes ranging from a minimum of 58 mm to a maximum of 410 mm caused by the 2007/6/3 *Mw* 6.4 (EQ4). Water level coseismic changes





reached 228 mm in response to epicentral distances greater than 3,000 km (EQ5). Water temperature induced

by earthquakes ranged from 2.1 to 24.5 Mk, where the maximum was caused by the 2011/3/24 *Mw* 7.2 (EQ7).

     Water levels also show the influence of Earth tides, with a maximum variation induced by semidiurnal and

diurnal tides being < 9 mm (Yang et al., 2005), which is less than the step-like changes induced by

earthquakes. Seasonal fluctuations in water level from rainy to dry seasons were < 200 mm and showed no

long-term trend. All earthquake-induced groundwater level responses were greater than 200mm, which

cannot be attributed to Earth solid tides and meteorological factors.

Table 1. Seismic properties for fourteen earthquakes and their effects on water-levels and temperatures in the Dazhai
Well, Western Yunnan Province, China.

| Earthquake ID | Date | Earthquake properties | | | | | Maximum change | | |
|---|---|---|---|---|---|---|---|---|---|
| | | Longitude (°E) | Latitude (°N) | Magnitude (*Mw*) | Epicentral distance (km) | Seismic energy density (J/m$^3$) | Water level (mm) | Discharge (L/s) | Temperature (mK) |
| EQ1 | Dec, 26 2004 | 95.79 | 3.05 | 9.2 | 2263 | 0.0941 | 380 | 0.168 | 6.1 |
| EQ2 | Mar, 29 2005 | 97.05 | 2.03 | 8.5 | 2344 | 0.0081 | 136 | 0.07 | 2.1 |
| EQ3 | May, 16 2007 | 100.96 | 20.57 | 6.6 | 240 | 0.0140 | 108 | 0.065 | 10.1 |
| EQ4 | Jun, 3 2007 | 101.13 | 23.08 | 6.4 | 36 | 2.2418 | 410 | 0.087 | 6.0 |
| EQ5 | Sep, 12 2007 | 101.41 | -4.9 | 8.6 | 3076 | 0.0050 | 228 | 0.05 | <0.1 |
| EQ6 | May, 12 2008 | 103.42 | 31.01 | 8 | 950 | 0.0235 | 279 | 0.122 | 6.2 |
| EQ7 | Mar, 24 2011 | 99.85 | 20.70 | 7.2 | 260 | 0.0817 | 257 | 0.133 | 24.5 |
| EQ8 | Apr, 11 2012 | 93.08 | 2.31 | 8.6 | 2430 | 0.0102 | 72 | 0.043 | 10.3 |
| EQ9 | Nov, 11 2012 | 95.91 | 22.88 | 7 | 510 | 0.0054 | 58 | 0.014 | 4.8 |
| EQ10 | Jan, 28 2014 | 101.17 | 22.51 | 4.6 | 30 | 0.0094 | 98 | 0.033 | 11.9 |
| EQ11 | Feb, 5 2014 | 101.01 | 22.80 | 4.2 | 4 | 1.1022 | 81 | 0.034 | 7.9 |
| EQ12 | Oct, 7 2014 | 100.50 | 23.40 | 6.6 | 100 | 0.1981 | 329 | 0.09 | 16.5 |
| EQ13 | Dec, 6 2014 | 100.49 | 23.32 | 5.8 | 90 | 0.0187 | 72 | 0.019 | 5.3 |
| EQ14 | Apr, 25 2015 | 84.65 | 28.15 | 8.2 | 1753 | 0.0072 | 146 | 0.049 | 13.1 |

Wang (2007) introduced the concept of seismic energy density to quantify the magnitude and epicentral

distance of earthquakes to build the relationship between seismic activity and hydrologic response, where a

threshold value of $10^{-3}$ J/m$^3$ was concluded to be sufficient to trigger a hydrological coseismic response. The

seismic energy density can be calculated by the following equation:

$$\log r \ = \ 0.48 \, Mw \ - \ 0.33 \, \log e \ - \ 1.4 \qquad\qquad (1)$$

where *r* is the epicentral distance, *Mw* is the magnitude of the earthquake and *e* (J/m$^3$) is the seismic energy

density. Table 1 summarizes the seismic energy density of the fourteen earthquakes, which ranged from 0.005

J/m$^3$ to 2.242 J/m$^3$. The detailed information is summarized in Table 1. The minimum seismic energy density

required to trigger water level and temperature coseismic changes were 0.0054 J/m$^3$ and 0.005 J/m$^3$,

respectively, where both values are > $10^{-3}$ J/m$^3$, approaching the empirical threshold for triggering hydraulic

property changes, which means that this well is highly sensitive to seismic activity. We next explore the

mechanism of these coseismic responses and their quantitative assessment, and then identify the origin of

excess recharge to the aquifer induced by earthquakes.

**4 Changes in Aquifer Parameters**

Aquifer parameters such as transmissivity, storage coefficient, and hydraulic diffusivity play important
roles in groundwater flow. During seismic activity, anomalous changes in water level may be attributed to
earthquake-induced changes in aquifer parameters. The tidal analysis method establishes a direct connection
between hydrological responses and aquifer parameter changes, which is also an effective method for the
long-term monitoring of variation in aquifer parameters (Hsieh et al., 1987; Xue et al., 2013; Shi and Wang,
2016). Water-level changes induced by Earth solid tides in a wellbore show semi-diurnal and diurnal
fluctuations, which are controlled by aquifer properties and the wellbore structure. The amplitude is the ratio
of the water level amplitude to the theoretical volumetric strain and the phase shift is the time lag between
the Earth solid tide fluctuation and water level variation, which differ because of the time required for
groundwater to equilibrate the well. Both can be used to calculate the transmissivity and storage coefficient:

$$X / H = (E^2 + F^2)^{-1/2} \tag{2}$$

$$\eta = -\tan^{-1}(F/E) \tag{3}$$

$$E \approx 1 - \frac{\omega r_c^2}{2T} Kei(\alpha), \ F \approx \frac{\omega r_c^2}{2T} Ker(\alpha), \ \alpha_w = (\frac{\omega S}{T})^{\frac{1}{2}} r_w \tag{4}$$

where $X/H$ is the amplitude, $X$ is the amplitude of tide of the water level in the well, $H$ is the amplitude of the
fluctuating pressure head in the elastic aquifer responding to the tidal stress, $\eta$ is the phase shift, $T$ is
transmissivity ($m^2$/d), $S$ is the storativity (dimensionless); $Ker$ and $Kei$ are zero-order Kelvin functions; $r_w$
and $r_c$ are the radii of the well and well casing, respectively, and $\omega$ is the tidal frequency (Hz).

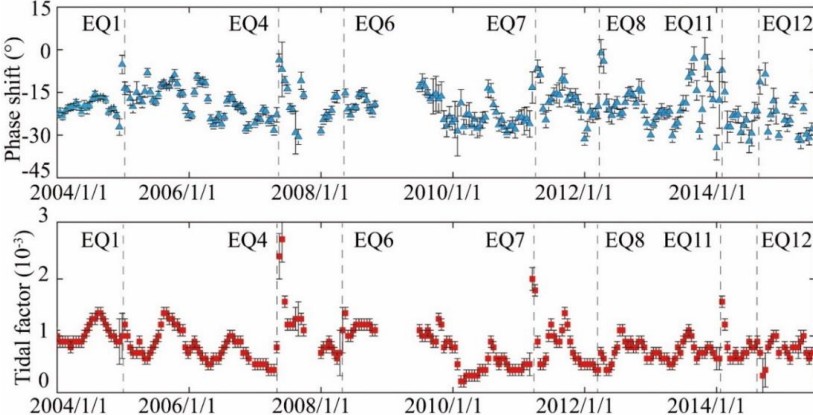

Figure 4. Phase shift and tidal factor of M2 tidal constituent using water-level observations in Dazhai Well between
2004 to 2015. Vertical dashed lines indicate earthquake times.

Spectral analysis was conducted to identify the multiple components inducing water level fluctuations in
DZ well (Figure S3 in the supporting information). The obvious amplitudes were observed in a frequency
band of 0~3cpd, indicating that the water level in Dazhai well responds to Earth tides (Qu et al., 2020). In
order to obtain the phase shift and amplitude information of the water level time series, we used tidal analysis
in the calculations. We used a common tidal analysis program (Baytap-G) and set a  period of 30 days with


an overlap of 15 days (Tamura et al., 1991). For Earth tide, $O_1$, $K_1$, $M_2$, and $S_2$ are the main waves that account for at least 95% of all tidal waves. Here, we only chose the $M_2$ wave for further analysis as it is stable and relatively unaffected by barometric pressure. We focus on the evolution in aquifer properties induced by seismic activity based on post-seismic water-level to recovery to the pre-seismic value that ranges from 13 to 70 days. We considered earthquakes that occurred within three months of each other as a single earthquake

to reduce post-seismic effects on the aquifer properties. During this time interval, earthquakes with the greatest seismic energy density were selected for tidal analysis, which totaled seven earthquakes (EQ1, EQ4, EQ6–EQ8, and EQ11–12). The three-month averages of phase shift and amplitude before and after earthquakes were used to calculate the aquifer parameters. The phase shift and tidal factor changes following earthquakes can be clearly identified in Figure 4, which are summarized in Table 2.

As is shown in Figure 4, we can find that the errors are much smaller than the changes that caused by the earthquakes, which represents the 95% confidence interval. There is no periodic variation on the tidal factor. During the monitoring period, the tidal factors range from $2 \times 10^{-4}$ to $1.4 \times 10^{-3}$, and the average value is $8 \times 10^{-4}$. During aseismic period the rate of changes ranges from 0 to 75% in comparing with the mean value of tidal factor. However, the changes are 237.5%, 87.5%, 150% and 100% following EQ4, EQ6, EQ7 and EQ11,

which are greater than that during aseismic period. Thus, the changes of tidal factor induced by earthquakes are real and significant.

Table 2. Aquifer parameters estimated using tidal analyses before and after selected earthquakes in Dazhai Well, western Yunnan, China

| | Date | Phase shift Change (°) | Amplitude Change (mm) | $S$ ($10^{-5}$) | $\triangle S$ ($10^{-5}$) | $T$ ($m^2$/d) | $\triangle T$ ($m^2$/d) | $D$ ($m^2$/s) | $\triangle D$ ($m^2$/s) | $\triangle D$ (%) |
|---|---|---|---|---|---|---|---|---|---|---|
| EQ1 | Dec, 26 2004 | 6.6 | 0 | 2.56 | 0 | 0.35 | 0.17 | 0.16 | 0.08 | 49 |
| EQ4 | Jun, 3 2007 | 8.8 | 3.7 | 4.67 | -3.20 | 0.32 | 0.3 | 0.08 | 0.41 | 515 |
| EQ6 | May, 12 2008 | -17.3 | 0.9 | 3.18 | -8.41 | 0.43 | -0.24 | 0.16 | 0.06 | 40 |
| EQ7 | Mar, 24 2011 | 13.8 | 1.8 | 4.67 | -2.40 | 0.28 | 0.46 | 0.07 | 0.31 | 440 |
| EQ8 | Apr, 11 2012 | 7.0 | 0 | 4.41 | 0 | 0.30 | 0.15 | 0.08 | 0.04 | 50 |
| EQ11 | Feb, 5 2014 | 5.5 | 1.5 | 3.45 | -1.36 | 0.43 | 0.24 | 0.14 | 0.23 | 157 |
| EQ12 | Oct, 7 2014 | 4.0 | -0.3 | 3.06 | 0.40 | 0.34 | 0.08 | 0.13 | 0.01 | 9 |

The average value of phase shift is -20° during 2004~2015, and it ranges from -31° to -11° during the period without earthquakes. However, the phase shifts following the earthquakes (except for EQ6, EQ11 and EQ12) are greater than -10°, which is also significant with that during aseismic period. During the pre-seismic and post-seismic period of EQ11 and EQ12, the phase shift shows a wide range of fluctuation which looks like 'noise', but it is not actually noise. The reason for the phenomenon may be that the epicentral distance

of EQ11 and EQ12 are 4km and 100km respectively, and the aquifer properties is continuous change caused by the crustal strain change during the earthquake preparation process. In addition, because of the short interval between the two consecutive earthquakes, the aquifer properties maybe effected by the previous earthquake and did not recover.

    As proposed by Hsieh et al. (1987), the phase shift and tidal amplitude are related to the aquifer

transmissivity and storage coefficient, so that changes in phase shift and tidal amplitude also indicate changes in aquifer hydraulic properties following earthquakes. The tidal amplitude reflects the ability of an aquifer



system to respond to Earth tidal changes induced by earthquakes and is affected by aquifer properties (e.g., undrained compressibility, Poisson's ratio, and fluid compressibility) while variations in phase shift are related to permeability changes (i.e., small phase shifts are related to high aquifer permeability, while large

ones are related to low permeability). The hydraulic diffusivity $D$ (m$^2$/s; equal to the ratio of $T$ to $S$), is introduced to accurately depict the changes in aquifer parameters, where results imply that earthquake cause the change of hydraulic diffusivity ranges from 9% to 515%.

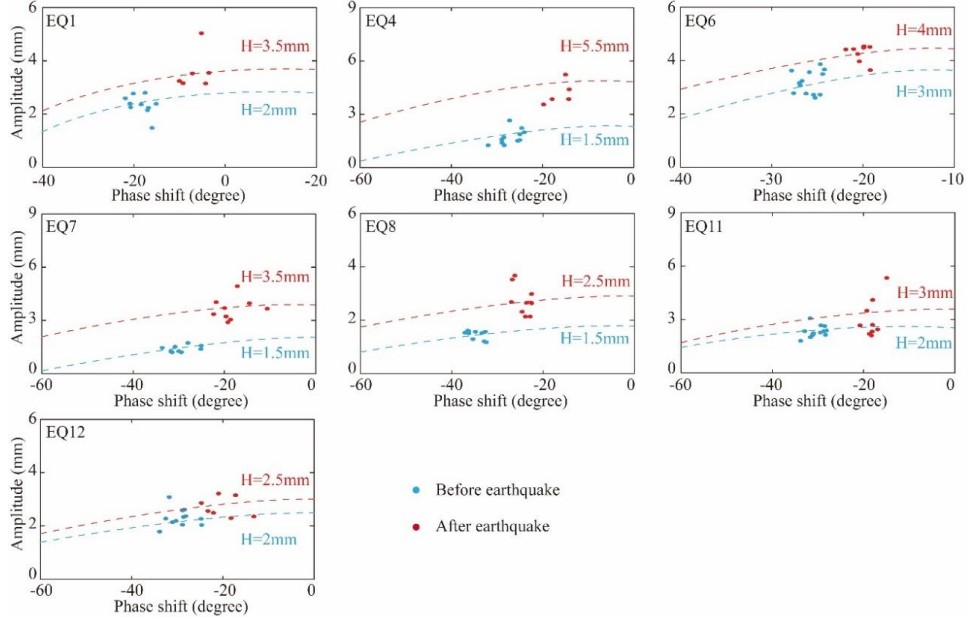

Figure 5. Amplitude versus phase shift at the frequency of the M2 wave. Dots before earthquake (blue dots) and after
earthquake (red dots) fit two curves that represent the theoretical model (equation 2 and 3) with different $H$, respectively.

The Hsieh's model only analyzes the earthquake-induced horizontal permeability changes. The vertical permeability change caused by earthquakes can be detected by comparing the amplitude and the tidal phase
shift (Liao et al., 2016; Wang et al., 2016). As is shown in Figure 5, the amplitude and phase shift inferred from before and after earthquake fall into two distinct groups (red dots and blue dots). In addition, for the given value of $H$, the $X$ versus $\eta$ relationship can be represented by a single curve in the plot, which is uniquely determined by $T$ and $S$. Dots before earthquake (blue) and after earthquake (red) fit two with different value of $H$, which indicated that the changes of vertical permeability in the aquifer system (Liao et al., 2016).

**5 Discussion**

**5.1 Mechanisms of Water-level Coseismic Responses**



Static and dynamic strain mechanisms have been proposed to explain coseismic water level responses where static strains change aquifer permeabilities resulting in variations in pore pressure and fracture aperture, while the effects of dynamic strain include 1) undrained consolidation and liquefaction, 2) drainage from the

unsaturated zone, 3) bubbles mobilization through highly fracture, 4) the formation of new fractures and 5) unclogging of preexisting fractures due to the propagation of seismic waves. In this section, we will discuss the impacts of static and dynamic strain on water level coseismic responses in DZ well.

### 5.1.1 Static Strain

Static strain induced by fault slip may lead to the dilate of fractures which will change the pore pressure

and the jog volume. For our data, static strains produced by large, distant earthquakes with epicentral distances >2000 km (EQ1 and EQ8) are lower than those earthquakes listed in Table 2, thus we use the near-field earthquakes listed in Table 2 to analyze the change of jog volume and fluid pressure.

The elastic or permanent deformation of crust associated with the fault slip cause static strains. It is assumed that the fault jog was occupied by the fluid before earthquakes. When the frictional strength of the

fault is exceeded under the accumulation of tectonic stress, we may estimate the relative volume change of fault jog according the seismic moment ($M_0$),

$$M_0 = \mu A s \tag{5}$$

where $\mu$ is the shear rigidity of the host rock, $A$ is the surface area of the rupture and $s$ is the average slip over the rupture area. The seismic moment can be converted to the moment magnitude ($Mw$),

$$Mw = 2/3 \log M_0 - 10.7 \tag{6}$$

The rupture length ($L$) of seismogenic fault is the function of moment magnitude (Wells and Coppersmith, 1994), as follow

$$\log L = 0.74 Mw - 3.55 \tag{7}$$

The down-dip width ($W$) and the average slip ($s$) can be calculated by the established empirical relation

$$W = L \qquad Mw < 4.0 \tag{8}$$

$$W = 15.0 L^{2/3} \qquad 4.0 < Mw < 6.0 \tag{9}$$

$$W = 17 km \qquad Mw > 6.0 \tag{10}$$

$$\log s = 0.833 \log L - 3.84 \tag{11}$$

Based on the above parameters, the relative volume change of fault jog could be estimated by the equation

listed below,

$$\triangle V = w D_j s \tag{12}$$

where $w$ is the vertical extents of the jog; $D_j$ is the jog length; It is assumed that $D_j$ =1m. The relative volume change of fault jog is summarized in Table 3 following different near-field earthquakes.

Based on the assumption of isothermal expansion, the fluid pressure changes in the fault jog associated

with the volume change of fault jog is considered as the behavior of ideal gas with equation of state $PV=nRT$





(Weatherley and Henley, 2013). According to the 11-year observed mean value of water level, the initial pressure ($P$) of the fluid before the earthquake are calculated as,

$$P(z) = \rho g z \tag{13}$$

Where $\rho = 1000$ kg/m$^3$, $g = 9.81$ m/s$^2$ is the gravitational acceleration, $z$ is the depth. Then, the relative change of fluid pressure is calculated by

$$\frac{P_f}{P_i} = \frac{V_i}{V_f} \tag{14}$$

where the subscripts '$i$' and '$f$' indicate values before (initial) and after (final) the earthquake. The significant different of final fluid pressure varying with the different depths are summarized in Table 3. For different earthquakes, the magnitudes of relative change of fluid pressure are different. According to the $\triangle P = \rho g h$, the relative change of fluid pressure can cause the magnitude of coseismic response of water level exceeding 80 meters in theory, which is inconsistent with the observed value of DZ well. Due to the action of discharge port on the wellbore, a part of coseismic increased groundwater is discharged by the discharge port. The actual maximum change of water level coseismic response should be greater than the observed value. We estimate that the actual maximum change of water level coseismic response should be 0.35 m~ 0.83 m which is the sum of the observed maximum change and the converting of volume of discharge groundwater into the height of water level in wellbore, basing on the equation of $V = \pi R^2 H$, where $V$ is the volume of discharged groundwater, $H$ is the height of water level, and $R$ is the radius of wellbore. But these results are still two orders of magnitude smaller than the theoretical value of water level change estimated by the relative change of fluid pressure. Thus, the water level coseismic response may not be attributed to the fluid pressure change induced by static strain.

Table 3. The relative change of volume and fluid pressure following the near-field earthquakes.

| Earthquake ID | $L$(m) | $W$(m) | $s$(m) | $V_i/V_f$ | Initial Pressure (MPa) | Final pressure (MPa) | The relative change of fluid pressure (MPa) |
|---|---|---|---|---|---|---|---|
| EQ4 | 1.53E+04 | 1.70E+04 | 1.41E-03 | 6.64E-02 | 8.41E-01 | 5.59E-02 | 0.7854 |
| EQ6 | 2.34E+05 | 1.70E+04 | 1.36E-02 | 7.29E-03 | 8.41E-01 | 8.19E-06 | 0.8413 |
| EQ7 | 6.00E+04 | 1.70E+04 | 4.38E-03 | 2.23E-02 | 8.41E-01 | 1.25E-03 | 0.8398 |
| EQ12 | 2.16E+04 | 1.70E+04 | 1.87E-03 | 5.08E-02 | 8.41E-01 | 1.30E-09 | 0.8405 |

The relative change of fluid pressure only calculates the theoretical value of relative magnitude of water level change, but cannot judge whether water level is rising or declining following earthquakes theoretically. The Okada (1992) model is introduced to further evaluate the relationship between the water level coseismic response and static strain, which indicates that the change of static strain cause the water levels decline in the expansion zone and rise in the contraction zone (Okada, 1992; Toda et al., 2013). Table S1 in supporting information summarizes fault-plane solutions obtained from the USGS for hypocentral depth, strike, dip and rake.

Observed coseismic water-level changes following EQ12 are consistent with this model, while the coseismic responses following EQ4 and EQ7 are opposite to the prediction (Figure S4 in supporting



information). We conclude that the distribution of coseismic water level responses during seismic activity is more complex than the quadrantal distribution expected from the theoretical model. Thus, basing on the analysis of the relative change of fluid pressure and the Okada model, the coseismic response of water level

in DZ well cannot be attributed to the static strain, because it only explains a subset of the coseismic water-level responses. In addition, the large distant earthquakes (epicentral distances >2000) also cause the coseismic responses of water level in DZ well, while static strain is extremely small for these distant earthquakes, thus implying that the mechanism of coseismic responses are attributed to dynamic strains.

### 5.1.2 Dynamic Strain

As DZ well is located in a highly fractured aquifer, the co-seismic hydrological responses are more likely caused by the mechanism of dynamic strain. Of the five potential dynamic causes, the first two (undrained consolidation and liquefaction, unsaturated zone drainage) can be omitted because the observation well is located within consolidated rocks and no liquefaction was observed during seismic activity. Moreover, the variation of water-level in DZ well is sensitive to Earth tides which indicates that the content of compressible

gaseous phase is less than $10^{-4}$% of the aquifer (Brodsky et al., 2003). Thus, the mechanism of bubbles mobilization through highly fracture is unlikely. In addition, the result of tidal analysis indicates that these earthquakes caused changes in aquifer properties; the phase shift increased following earthquakes and recovered within several weeks to months afterward, indicating that the earthquake-induced permeability changes were transient, not permanent. The permeability and poroelastic properties change within the fault

damage zone may be attributed to the dynamic strain caused by the passage of seismic waves. Thus, we consider that earthquake-induced dynamic stress leading to unclogging of pre-existed fractures under the propagation of seismic waves is a plausible mechanism for the observed behavior. According to the aquifer lithology of Dazhai deep well, there existed two different aquifers which is separated by a thin aquitard (mudstone), and the pore pressure of the deeper aquifer is higher than the shallow one, such hydrogeological

setting makes the upwelling of groundwater from deep aquifer possible when the hydraulic connection is created following earthquakes (Figure 6). In the unclogging process, precipitates and colloidal particles are removed from the clogged fractures by the propagation of seismic waves, which connects aquifers at different depth. After earthquakes, permeabilities gradually recover through geochemical and biogeochemical processes or by the reclogging of fractures, causing water levels to recover to pre-earthquake levels (Manga

et al., 2012; Xue et al., 2013; Shi and Wang, 2015).

    Yet, unclogging only qualitatively explains water-levels change following earthquakes and does not explain the different magnitudes of changes following different earthquakes. Alternatively we infer that the different magnitudes of water level change are associated with the total volume of excess recharge to the aquifer during seismic activity, where the volume of aquifer recharge can be described by (Wang et al., 2004):


$$q = \frac{8Q}{\pi^2 \tau} \sum_{n=0}^{\infty} \exp\left[ -(2n+1)^2 \frac{t}{\tau} \right] \tag{15}$$



where $Q$ is the total excess water recharge to the aquifer from deeper aquifer, $t$ is the time since the earthquake occurred, and $\tau$ is characteristic time. The solution to the groundwater flow equation can be determined by two parameters ($Q$ and $\tau$) that are determined by fitting Equation (15) to the observed discharge time series using the non-linear least-squares Marquardt-Levenberg algorithm (Figure 7). The accuracy and efficiency of model fitting results are shown in Table S2 and Figure S5 of the supporting information.

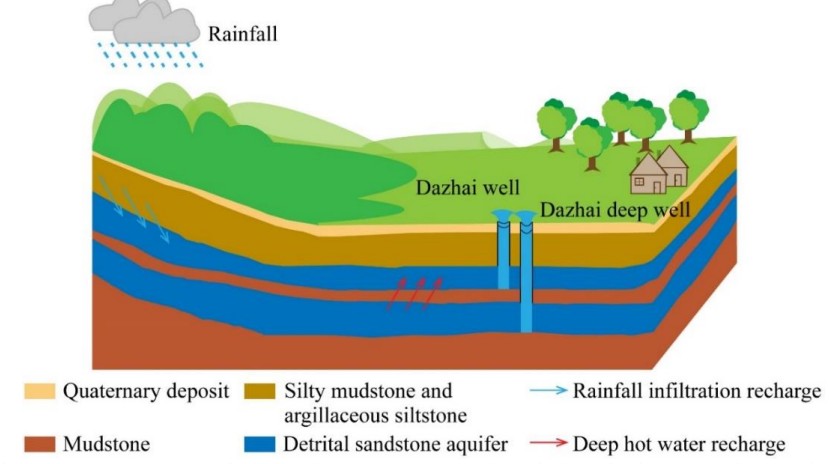

Figure 6. Conceptual model for the recharge process and hydrological coseismic response in Dazhai well.

Figure 7. Model results compared with normalized discharge data for the discharge coseismic response in DZ well.

Excess recharge predicted by the one-dimensional conceptual model are summarized in Table 4. The excess recharge caused by earthquakes differs for different earthquakes within the range of 85–273 m³. We





conclude that this is the most plausible reason for the variation in water-level change magnitudes following
each earthquake.

Table 4. Parameters estimated by the data of discharge coseismic response in Dazhai Well,
Western Yunnan Province, China.

| Earthquake ID | Date | $Q$ (m$^3$) | $\tau$ |
|---|---|---|---|
| EQ1 | Dec, 26 2004 | 265 | 2917 |
| EQ4 | Jun, 3 2007 | 273 | 2168 |
| EQ6 | May, 12 2008 | 85 | 393 |
| EQ7 | Mar, 24 2011 | 204 | 836 |
| EQ8 | Apr, 11 2012 | 101 | 663 |
| EQ11 | Feb, 5 2014 | 87 | 510 |
| EQ12 | Oct, 7 2014 | 96 | 605 |

**5.2 Possible Mechanism of Water Temperature Coseismic Change**

The dynamic strain mechanism describes the process of coseismic water level change, while the one-
dimensional diffusion model can explain the different magnitudes of anomalous water level change following
each earthquake. Because they cannot reveal the source of water recharge to the aquifer, we analyzed the co-
seismic water temperature changes to further identify the origin and processes of the coseismic groundwater
responses. Several mechanisms have been proposed for groundwater temperature changes following
earthquakes, including heat loss through the walls of boreholes (Kitagawa and Koizumi, 2000), gas escape
from groundwater (Ma, 2016; He et al., 2017), friction induced by seismic waves (Fulton et al., 2013; Tanaka
et al., 2006; Mori et al., 2010) and changes in the mixing ratios of groundwaters with different temperatures
(Shi et al., 2018; Mogi et al., 1989). Although heat can be propagated by radiation, convection through pores,
and conduction through solid phases, solar radiation only occurs at the earth's surface, which has little effect
on groundwater temperature. Also, temperatures should change gradually if groundwater exchanges heat with
the rock matrix, which is inconsistent with our observations. The gas-escape mechanism explains
groundwater temperature changes caused by large amounts of dissolved gasses escape from groundwater (a
phase-change from liquid to gas) that absorbs heat from the surrounding environment that is then dissipated,
causing the water temperature to decrease. Yet this explanation is also inconsistent with our step-like
increases. Friction heat caused by seismogenic faults may explain co-seismic water temperature responses in
some locations close to earthquake epicenters, but cannot cause temperature responses at great distances.

Because groundwater is an important carrier fluid for heat transfer, simultaneous change in water level,
flow rate, and temperature occur simultaneously, which indicates that groundwater flow plays an important
role in coseismic responses water temperature for our system. Thus, the mechanism that mixing of different
sources of water is likely to explain the synchronous changes in groundwater discharge and temperature.
Several previous studies have documented that mixing of deeper fluids and meteoric water in the crust may
play a key role in changing the physical and chemical characteristic of the groundwater (Kaown et al., 2021;
Li et al., 2019; Skelton et al., 2014b; Jellicoe et al., 2021; Woith et al., 2013). Fault zone could be acting as
a good conduit that connect the shallow and deep groundwater. For the hydrothermal spring in fault zone, it
is usually mixing with the shallow cold water during its rising to the surface. And groundwater in fault zone





aquifer may also origins from different depth of the aquifers. Previous studies suggested that earthquake could lead to the enhancement of permeability in the low permeability layer (Fischer et al., 2017) or reduction of permeability in high flow path (Shi et al., 2018), causing the changes of mixing ratio between shallow and deep groundwater, and thus lead to the changes of flow rate and chemical components in groundwater (Sato et al., 2020; Li et al., 2019). However, few studies had investigated the earthquake-induced changes both in

discharge and temperature in a quantitative way. End-member mixing model is widely used to quantify the differential contributions of groundwater from two or more sources, which is based on assumptions that the spatial distribution of groundwater with distinct characteristics is nonuniformity. In this study, we use the end-member mixing model to describe the anomalous change of discharge and water temperature following earthquakes. The possible end-member is analyzed and identified in the following part.

The climatic characteristics in the Simao basin is moderate climate with four distinct seasons. The annual precipitation is about 1500 mm, but most precipitation occurs between April and October. The stable-isotope data indicates that the recharge of DZ well aquifer is primarily from meteoric waters. We observed an increase of the water level and flow rates which are not related with precipitation as only 3mm and 0.9mm occurred in the December, 2004 (EQ1) and February, 2014 (EQ11), respectively. Wavelet coherence analysis is widely

used to identify the influences on water temperature variation in the absence of earthquakes (Zhang et al., 2020). As shown in the Figure S6 of Supporting information, there is high coherence between meteorological and hydrological data for around a year throughout the measurement period. However, the mean phase angles between discharge (water temperature) and rainfall are about 45°, indicating a lead of rainfall about 1-1.5 years, which implies that the excess recharge to the aquifer is not directly originated from precipitation.

Although the DZ Well only penetrate one aquifer, according to the borehole information of Dazhai Deep Well, there existed two different aquifers which is separated by a thin mudstone aquitard. Seismic activities cause the increased permeability of aquitard and make the upwelling of groundwater from deep aquifer possible when the hydraulic connection is created (Figure 6). Thus, we consider the mixing process between them.

For the selection of end-member, there are no limitations on the number of them in the hydrologic model, but the general rule is to use as few as possible to describe the observed anomalous phenomena (Laaksoharju et al., 2008). Thus, we assume that the different magnitudes of water temperature change are also caused by the mixing of these from distinct end member at different ratios following earthquakes. the earthquake-induced mixing process between the shallow and deep aquifer can be described by the following equations

(Kitagawa and Koizumi, 2000):

$$D_{\text{observed}} = D_{\text{s}} + D_{\text{d}} \qquad\qquad (16)$$

$$T_{\text{observed}} = \frac{T_{\text{s}} D_{\text{s}} + T_{\text{d}} D_{\text{d}}}{D_{\text{s}} + D_{\text{d}}} \qquad\qquad (17)$$

where $D_{\text{observed}}$ is the observed discharge, $D_{\text{s}}$ and $D_{\text{d}}$ are the discharges from shallow and deep aquifer, respectively; $T_{\text{observed}}$ is the observed water temperature, and $T_{\text{s}}$ and $T_{\text{d}}$ are the water temperatures of shallow

and deep aquifer, respectively. We infer that the water temperature responses caused by earthquakes are related to changes in discharges from the shallow and deep aquifer.

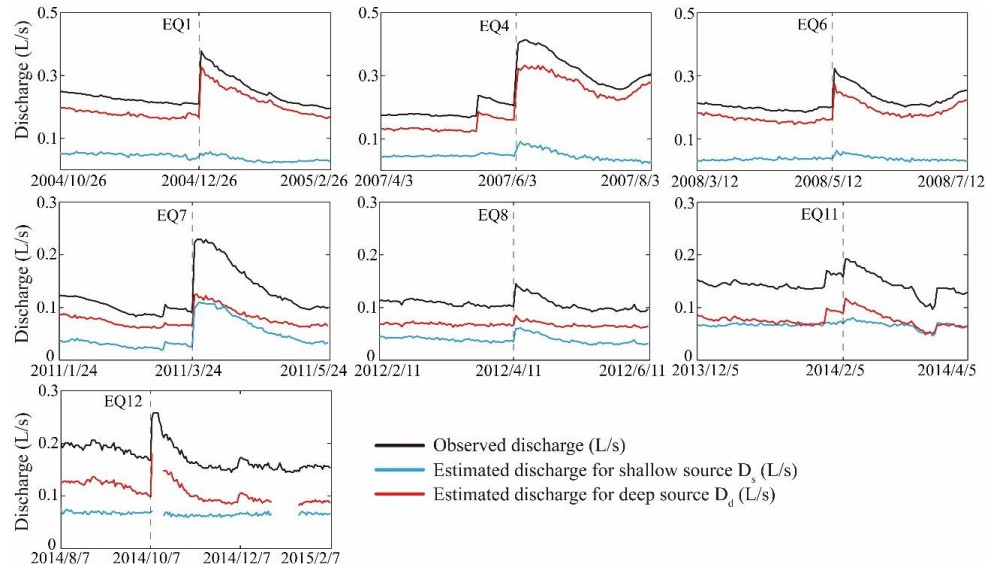

Figure 8. Observed and estimated discharge for shallow and deep source following selected earthquakes.

During the monitoring period, temperatures ranged between 21.7788 and 21.9325 °C, a difference of 153.7 mK that remained within this range even following seismic events. To explain temperature variation following these events, we employed a two-end-member mixing model where we assume two aquifers at different depth that are hydraulically connected following earthquakes: one is shallower with a lower temperature (21.7788 °C), while the other is deeper with a higher temperature (21.9325 °C) that correspond

to the extremal temperatures observed during the period of record.

Discharges from each end-member are then estimated using:

$$D_s = D_{observed}\ \frac{T_{observed} - T_d}{T_s - T_d} \tag{18}$$

$$D_d = D_{observed}\ (1 - \frac{T_{observed} - T_d}{T_s - T_d}) \tag{19}$$

Figure 8 presents the calculated discharges for each end member, with End-members 1 and 2 changing

following earthquakes; for EQ1, EQ11, and EQ12, the estimated discharge for shallow aquifer remained unchanged before and after earthquakes while discharge from deep aquifer increased sharply after earthquakes that may be due to the permeability increase in deep aquifer, which results in more warm groundwater entering the wellbore. For the other four earthquakes, the estimated discharges from shallow and deep aquifers varied, with the magnitude of change varying according to the earthquake. The change in

discharge from deep aquifers was always greater than that from shallow aquifers, which is the main cause of the coseismic water temperature response. Thus, we conclude that the coseismic water temperature response

can be attributed to earthquake-induced permeability increases, which result in warmer groundwater entering the wellbore. The different magnitudes of water temperature change following each earthquake are caused by changes in the mixing ratios of shallow and deep aquifers with different temperatures.

**6 Conclusion**

We documented groundwater levels, temperatures, and flow-rate responses following fourteen earthquakes in an artesian well in western Yunnan Province, China. The well always shows step-like groundwater level responses and increased flow rates and temperatures following earthquakes. Tidal analysis indicates that the aquifer's hydraulic diffusivity changes following earthquakes. Due to unclog preexisting fractures under the
propagation of seismic waves, hydrogeological setting makes the upwelling of groundwater from deep aquifer possible when the hydraulic connection is created following earthquakes, which is the likely mechanism explaining the coseismic water level response. The excess groundwater recharge from the deep aquifer, ranging from 85 m$^3$ to 273 m$^3$ for the fourteen earthquakes, leads to different magnitudes of water level changes. A groundwater mixing model based on observed water temperatures is used to estimate mixing
processes of waters with different temperatures before and after earthquakes, which shows that earthquakes induce recharge from both shallow and deep aquifers.

**Author contribution**

Shouchuan Zhang: Methodology, Data curation, Formal analysis, Writing – original draft. Zheming Shi: Conceptualization, Formal analysis, Writing – original draft, Writing – review & editing. Guangcai Wang:
Writing – review & editing, Funding acquisition. Zuochen Zhang: Data curation, Writing – review & editing.

**Competing interests**

The authors declare that they have no conflict of interest.

**Acknowledgements**

This work was supported by the National Natural Science Foundation of China (U1602233, 41972251). The
data used in this study are available from the database of Zenodo (https://zenodo.org/record/4746408#.YJlW64fis2x).

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
