# Peer review of "The origin of hydrological responses following earthquakes in confined aquifer: Insight from water level, flow rate and temperature observations"

_Hydrology and Earth System Sciences, 2022_

## Author Response (AR1)

Dear editor,

Thanks for the helpful comments provided by you and the reviewers, we have made necessary revision accordingly. The detail response could be found as follow:

Sincerely,
Zheming Shi

**Reviewer 1:**

Authors have analyzed water level, flow rate, and water temperature data from an artesian well in southwestern China before and after multiple earthquakes. Water level and temperature showed co-seismic step-like increases following earthquakes. Tidal analysis revealed changes in aquifer and aquitard permeability following earthquakes. Authors coupled the flow rate and temperature data to model the mixing processes that occurred following each earthquake. Results indicate that co-seismic temperature changes are the result of the mixing of different volumes of water from shallow and deep aquifers, with the mixing ratio varying according to each earthquake. I think the manuscript is interesting and suitable for HESS. Resent manuscript should be soon published. Authors thought useful not to consider atmospheric pressure since not monitored at the monitoring site, anyway barometric fields are characterized by large extensions. Authors may decide to ask data to Simao Airport or no, just to evidence relevant eventual variations. In any case present tractation may be considered suitable and convincing.

**Response:** We collect the monitoring data of barometric pressure in Simao City. In order to identify the effect of barometric pressure on the variation of water level in Dazhai well, the wavelet coherence analysis is employed to explore the correlation between water level and barometric pressure, and then the tidal components of water level in Dazhai well which extracted from the water level signal is compared under and without the influence of barometric pressure. Taking EQ1 as an example to analyze. The hourly monitoring data of water level and barometric pressure from Sep, 1 2004 to May, 30 2005 are chosen for the wavelet coherence analysis and extracting the tidal

components.

The wavelet coherence is used to explore the relationship between water level response to barometric pressure, which is a powerful tool to analyzing nonstationary signals. We convert the time series of water level and barometric pressure into time-frequency space based on the wavelet coherence. In Figure R1, the water level and barometric pressure are highly correlated at a 95% pointwise confidence level with coherence coefficients > 0.9 within band between 0.5 and 1 day. The semidiurnal period is evident throughout the entire data set. Periods of approximately 1 day are slightly unstable. The result of wavelet analysis indicates that the variation of water level is affected by barometric pressure.

[Figure]

Figure R1. Wavelet coherence between water level and barometric pressure. The thick black contour specifies the 95% confidence level. The arrow directions indicate the relative phase relationship: in-phase pointing right, antiphase pointing left, and phase-leading by 90° pointing straight down.

We compare the tidal components extracted from the water level time series under and without the influence of barometric pressure. Baytap-G program is used to remove the interference caused by barometric pressure in the time series of water level (Tamura et al., 1991). The time series can be divided into several items, as follow:

$$y_i = \sum_{m=1}^{M}\left(\alpha_m C_{mi} + \beta_m S_{mi}\right) + \sum_{k=0}^{K} b_k x_{i-k} + d_i + e_i$$

(R1)

Where the first term and second term on the right-hand side represent the tidal components and barometric pressure components, respectively; $d_i$ is the long-time drift;

and $e_i$ is the random noise. By using Baytap-G program, the tidal components are extracted from the water level time series with the elimination of barometric pressure (Figure R2). Compared the tidal components extracted from the water level time series under and without the influence of barometric pressure (Figure R2 and R3), the results are similar and it is indicated the effect of barometric pressure on the variation of water level has little effect on the extraction of tidal components from the water level time series.

[Figure]

Figure R2. Tidal analysis after removing barometric pressure by Baytap-G (a) Amplitude ratio and (b) Phase shift. The gray dash line indicates the time the earthquake occurred.

[Figure]

Figure R3. Tidal analysis after removing barometric pressure by Baytap-G (a) Amplitude ratio and (b) Phase shift. The gray dash line indicates the time the earthquake occurred.

**References:**

Tamura, Y., Sato, T., Ooe, M., Ishiguro, M. A procedure for tidal analysis with a Bayesian information criterion. Geophysical Journal International, 104(3): 507-516, https://doi.org/10.1111/j.1365246X.1991.tb05697.x, 1991.

**Reviewer 2:**

Comments on HESS-2002-326:

The paper by Zhang and others was trying to investigate the origin of hydrological responses following earthquakes in confined aquifer by monitoring the variations in groundwater levels, flow rate, and temperature. The topic of this paper is novel, interesting, and suitable for this journal. It was well written with good organized structure. There are still some issues that need to be modified.

Specific remarks:

1. Line 16: "the post-seismic discharge of 85~273 m$^3$." Is this daily or hourly discharge?

**Response:** The calculation results is obtained from fitting the monitoring data of flow rates for 20 days after earthquakes. Thus, the post-seismic discharge of 85~273 m$^3$ is the total excess water recharge to the aquifer from deeper aquifer in 20 days after

earthquakes. We have revised it in Line 18 of manuscript.

2.The authors state that there is no barometric pressure record in the well, but could it be possible to get the barometric pressure from nearby place? Sometime the barometric pressure may have large impact on the water level fluctuation.

**Response:** As response to Reviewer 1#, we collected the monitoring data of barometric pressure in Simao City, and identified the effect of barometric pressure on the variation of water level by the wavelet coherence and tidal analysis. Taking EQ1 as an example to analyze. The result of wavelet coherence indicates that the variation of water level is affected by barometric pressure. The result of tidal analysis indicates that the tidal components extracted from the water level time series under and without the influence of barometric pressure are similar. Thus, the new analysis supports our idea that barometric pressure fluctuation would not have effect on the result of our tidal analysis.

We have added the discussion in Line 220~222 of manuscript and Text S6 in the supporting information.

3.Temperature fluctuations seem to be affected by air temperature (Figure 3). Thus, I think that the effect of seismic activities on temperature may be more remarkable by removing the interference of air temperature with water temperature.

**Response:** Wavelet coherence is employed to analyze the correlation between water temperature and air temperature in time domain and frequency domain. Taking EQ1 as an example to analyze. The result of wavelet coherence indicates that there is no correlation between water temperature and air temperature (Figure R4). In addition, the temperature probe is located about 100m below the water surface, which is helpful for removing the influence of air temperature on the variation of water temperature. Thus, air temperature fluctuation would not have effect on the variation of water temperature. We have added the discussion in Line 146 of manuscript and Text S3 in the supporting information.

[Figure]

Figure R4. Wavelet coherence between water temperature and air temperature. The thick black contour specifies the 95% confidence level. The arrow directions indicate the relative phase relationship: in-phase pointing right, antiphase pointing left, and phase-leading by 90° pointing straight down.

4.What is the different effect of static and dynamic water level observation on the tidal signal? Why dynamic water level observation would show more sensitive to earthquake? Please give a detailed explanation

**Response:** We have added the discussion in Line 103~111 of revised manuscript.

For the earthquake groundwater monitoring, the wells tapped in the confined aquifers are preferred. The static water level observation is used for the non-artesian well, while the dynamic water-level observation is used for the artesian well tapped in well-confined aquifers. Thus, the dynamic water level observation wells would tend to have better confinement than the non-artesian flowing wells. Previous studies revealed that three external influence factors affect the sensitivity of the earthquake-induced hydrogeological responses, including peak ground velocity (PGV), aquifer confinement, and well location relative to local faults (Zhang et al., 2021). In additional, many studies have proposed that the tidal signals from aquifers with different confinement are different. The aquifers with better confinement could record clear and large magnitude of $M_2$ tidal wave while the aquifer with less confined may show weak tidal signals (Rahi, 2010; Turnadge et al., 2019; Zhang et al., 2021). Thus, dynamic water level observation tapped would tend to show more sensitive to earthquake.

5.Line 112 the authors mentioned that there is stable isotope result from this well, could this result be used to estimate the recharge elevation? If this is feasible, this may provide additional support of the concept model proposed in the manuscript.

**Response:** We have added the discussion in Line 128~133 of revised manuscript.

The $\delta D$ and $\delta^{18}O$ values of groundwater from Dazhai well are -80.63‰ and -11.17‰, respectively, which are close to the global meteoric water line GMWL: $\delta D=8\delta^{18}O + 10$. The result of stable-isotope indicate recharge is primarily from meteoric waters. According to the relationship between recharge elevation and $\delta D$ in Eastern Tibetan Plateau $\delta D = -0.026 H(m) - 30.2$ (Yu, 1997), the recharge elevation of groundwater from Dazhai well is about 1940m. The estimated recharge elevation is close to the elevation of mountains nearby Dazahi well.

6.The authors selected 14 earthquakes that cause co-seismic response, and discussed the relationship between seismic energy density limits and the response, here I am wondering whether the authors could also plot those earthquakes that showed no response, and see if there is any limits exist between the response and non-response earthquakes?

**Response:** We have added the discussion in Line 175~178 of revised manuscript.

We have plotted those earthquakes that showed hydrological response and no hydrological response in Figure R5. Orange triangles represent earthquakes with hydrological response collected from Wang and Manga (2010). Blue squares represent 14 earthquakes in this study that induce hydrological response in DZ well. White circles represent earthquakes without hydrological response in DZ well.

Wang and Manga (2010) collected and analyzed the global earthquake-induced hydrological responses, and found that the seismic energy density $> 10^{-3}$ J/m$^3$ can induce hydrological response. As is shown in Fig R5, although the seismic energy density of most earthquakes are $< 10^{-3}$ J/m$^3$, the seismic energy density of few earthquakes ranging from $10^{-3}$ J/m$^3$ to $10^{-2}$ J/m$^3$ cause no hydrological response. The

seismic energy density of 14 earthquakes with hydrological response are $> 10^{-3}$ J/m$^3$ (ranging from 0.005 J/m$^3$ to 2.242 J/m$^3$).

[Figure]

Figure R5. Distribution of earthquake-triggered hydrologic changes as a function of earthquake magnitude and distance. Orange triangles represent earthquakes with hydrological response collected from Wang and Manga (2010). Blue squares represent 14 earthquakes of this study that induce hydrological response in DZ well. White circles represent earthquakes without hydrological response in DZ well.

7. As for the static strain mechanism, there are many parameters such as W, L, Dj…, how these values are determined? If these are empirical, then citation should be added.

**Response:** We have added the references of empirical value and assumption value in the static strain mechanism. (Line 286-287, Line 290, Line 298-299)

8.Have the authors sampled the water samples from the shallow and deep aquifers? Is there any difference in hydrochemical components of groundwater from the two aquifers?

**Response:** We have added the discussion in Line 95~97 of revised manuscript.

We have sampled groundwater from Dazhai well and Dazhai deep well. The hydrochemical type of Dazhai well and Dazhai deep well are HCO$_3$-Na·Ca and HCO$_3$-Ca· Na, respectively. There is no significant difference in groundwater hydrochemical type which are sampled from the shallow and deep aquifer.

Table R1. The ion concentration of groundwater samples.

| K$^+$ (mg/L) | Na$^+$ (mg/L) | Ca$^{2+}$ (mg/L) | Mg$^{2+}$ (mg/L) | HCO$_3^-$ (mg/L) | NO$_3^-$ (mg/L) | Cl$^-$ (mg/L) | SO$_4^{2-}$ (mg/L) |
|---|---|---|---|---|---|---|---|

| | | | | | | | | |
|---|---|---|---|---|---|---|---|---|
| DZ Well | 1.26 | 47.15 | 37.92 | 6.74 | 267.21 | 0.59 | 0.94 | 9.98 |
| DZ deep well | 1.66 | 42.84 | 40.99 | 10.58 | 288.632 | 0.41 | 0.7 | 6.49 |

9. Line 136: What does 'Mk' mean?

**Response:** mK means millikelvin. 1 mK is equal to 0.001℃.

**References:**

Bredehoeft, J.D. Response of well-aquifer systems to Earth tides. Journal of Geophysical Research (1896-1977), 72(12): 3075-3087. DOI:https://doi.org/10.1029/JZ072i012p03075, 1967.

Rahi, K.A., 2010. Estimating the hydraulic parameters of the Arbuckle-Simpson aquifer by analysis of naturally-induced stresses, Ph.D. dissertation. Oklahoma State University.

Turnadge, C., Crosbie, R.S., Barron, O., Rau, G.C. Comparing Methods of Barometric Efficiency Characterization for Specific Storage Estimation. Groundwater, 57(6): 844-859. DOI:https://doi.org/10.1111/gwat.12923, 2019.

Wang, C.Y., Manga, M. Hydrologic responses to earthquakes and a general metric. Geofluids, 10(1‐2). DOI:https://doi.org/10.1111/j.1468-8123.2009.00270.x, 2010.

Yu, J., 1997. Isotopic Geochemistry in China. Chinese Science Press.

Zhang, H. et al. Different Sensitivities of Earthquake-Induced Water Level and Hydrogeological Property Variations in Two Aquifer Systems. Water Resources Research, 57(5): e2020WR028217. DOI:https://doi.org/10.1029/2020WR028217, 2021.

**Comments by Community:**

This review was prepared as part of graduate program Earth & Environment (course Integrated Topics in Earth & Environment) at Wageningen University, and has been produced under supervision of dr Ryan Teuling. The review has been posted because of its potential usefulness to the authors and editor. Although it has the format of a regular review as was requested by the course, this review was not solicited by the journal, and should be seen as a regular comment. We leave it up to the author's and editor which points will be addressed.

Dear Joshua Leusink,

Thanks for the detail and helpful comments, we have read it carefully and made

necessary. This reply is following your point by point comments: And the detial response could be found as follow.

General comments

Single well

To start with issue (1): in the research only one single well was used to retrieve the data from. The other deeper well is not used in the research as an input location for the main research results and data acquisition. Therefore, the obtained results may either only explain the local hydrological responses or the hydrological response is the result of an overlooked factor. Therefore, one could argue how significantly and plausible the results represent the area and the processes taking place in the aquifers. Using a larger number of wells will cover the possible local bias in the results. Therefore, the regional effect can be described better (Mohr et al. 2019, Cox et al. 2012, Hosono et al. 2019). Another possibility could be to use discharge data coming from springs in the area, this was also used in the research of Manga and Rowland (2009). Although the discharge in a spring could be different from the increased discharge in a well, this extra data source could still enhance the significance of the dataset. Of course, temperature data could be a bit ambiguous, I would not advice to use this spring temperature.

**Response:** Firstly, it would of course be good that if we could have more data and wells involved in the study. However, only two wells (Dazhai well and Dazhai deep well) could be collected in this small basin. And Dazhai well and Dazhai deep well are monitored for different items. Dazhai well is used to conducted hydrogeological monitoring, including water level, water temperature and flow rates. Dazhai deep well is used to conduct geophysical monitoring, which is equipped with clinometer, seismometer, magnetometer, and strain gauge. Thus, only Dazhai well has continously monitoring hydrological data and can be used to analyze the hydrogeological response to earthquakes.

The hot springs and saline springs are mainly distributed along the faults in the northern part of Lanping—Simao basin, while few springs occur in the southern part of this basin. However, there is no continuously measurement of these springs. In addition,

Dazhai well is located in the Yixiangba sub-basin which belongs to the Lanping-Siamo basin. In this secondary hydrogeological units, there is no hot spring. Thus, it is with regret that no long-term monitoring data from other springs nearby Dazhai well can be used to analyze the hydrogeological coseismic response.

Research focus and motivation

Following with issue (2): in the section 'Geological setting and data sources', which could be acknowledged as the "Site description", the authors explain about the geological setting of the whole basin, to my opinion this also influences the resulting hydrological response in the aquifer of the well. However, in the conclusion and the other parts of the research, this broader perspective has not been touched upon.

Therefore, it would be recommended to either point out that only the Dazhai well and the aquifers it is directly connected to are of importance, or one should broaden the perspective to the whole (or part of the) basin. I would suggest to either make clear that this research only has investigated the Dazhai well (Most straight forward). Or discuss in the discussion section about the representability of this single well to the whole area/aquifer. This can be done with extra boreholes to gain extra lithostratigraphic knowledge of the area or by using the already provided information about the present aquifers.

**Response:** Thanks for your helpful suggestion. The Dazhai well-aquifer system is the most important research object in this study. We only focus on the coseismic hydrological response of Dazhai well-aquifer system instead of the whole basin or sub-basin. We have adjusted the paragraph structure of Chapter 2 and added the secondary title. The geological setting of the whole basin is considered as the background knowledge and prerequisite of coseismic hydrogeological responses.

Statistical analysis

Following with issue (3): in the research the statistical analysis is not always consistent and is sometimes not present. The authors make use of three models, the Hsieh model,

Marquardt-Levenberg algorithm and the end-member mixing model. They provided clear and good overview figures (figure 5: Hseih's model, figure 7: Marquardt-Levenberg algorithm and figure 8: End-member mixing model) of the obtained results and the model outputs. However, without any significance test or model and observation tests it doesn't make sense what the figures represent. For me it is hard to discuss whether the observations are well described by the model or not. The figures can give a wrong visualisation of the data. For the Marquardt-Levenberg algorithm this has already been provided in the supplements. Suggested would be, to provide a more or less equal type of statistical analysis that already have been done for the Marquardt-Levenberg algorithm (in supplement S5). Provide in this test how well the model represents the observations.

**Response:** For the Hsieh's model, the phase shift and tidal factor inferred from water level by Baytap-G have calculation errors. Actually, the significance test of results has been calculated in the Hsieh's model. The error bars in Figure 5 represent the 95% confidence interval which indicates that there is 95% probability of occurrence for calculation results (phase shift and tidal factor) within the range of confidence interval.

The most essential difference between Marquardt-Levenberg algorithm and end-member mixing model is that Marquardt-Levenberg algorithm is used to fit to the observed values, which can be conducted for statistical analysis between the fitted values and monitored values, while the results of end-member mixing model is not from the fitting value of monitoring values but from the calculation results of equation 18 and 19. Thus, it is hard to conduct the statistical analysis for calculation results.

Research question(s) and hypotheses

Concluding with issue (4): The paper does not indicate what the research question is. In the introduction it stated what the investigation will be about, but it does not mention what question(s) will be answered. Based on the introduction and abstract it becomes clear what the aim of the research is. Due to the lack of this research question, it is hard to check whether the research question(s) are answered within the research and conclusion.

Advised is to provide one or more research questions. Preferably with hypothesis to strengthen the research. The goal of a hypothesis is that a provided research question is hypothetically answered and could be either rejected, adjusted, or approved based on the discussion of the results.

**Response:** Thanks for your helpful suggestion. We have revised the manuscript in Line 12~14 and Line 40~43, and indicated research question in the abstract and introduction section, as follow:

"In this study, we quantitatively analyze the mechanism of coseismic response in water level and flow rate from an artesian well in southwestern China before and after multiple earthquakes, and investigate the origin of the earthquake-induced hydrological response based on the monitoring data of water temperature."

"Based on the coseismic response of water levels, flow rate and water temperature responses to several earthquakes in an artesian well (Dazhai well), this study aims to: (1) quantitatively evaluate the mechanism of anomalous hydrological changes. (2) investigate the origin of the earthquake-induced hydrological response based on water temperature monitoring data."

Minor comments:

1. Research paper structure: Paper misses a clear Methodology and results section. In the manuscript I can find where these sections are, but it is not clear at first sight. Moreover, it seems that in the result section and even the discussion new methods are introduced; Like the end-member mixing model (line 385). This model is an important model that is needed to find the answer to the research question. Consider moving this to a Methodology section.

**Response:** Disagree. In this study, several methods are used to analyze the hydrological coseismic phenomena and reveal the mechanisms of coseismic hydrological response, including the estimation of jog volume and fluid pressure change, tidal analysis, Okada model, the estimation of volume of aquifer excess recharge, and end-member mixing model. Different methods correspond to different mechanisms. If these methods are introduced and summarized in the methodology chapter, it may confuse the readers to

find exactly which method corresponds to each mechanism. Thus, the methodology section is not added in the revised manuscript. To make readers to find methods clearly corresponding to the different mechanisms, we have added the flow chart of discussion section in the supporting information (Figure R6).

In order to make the content of each chapter in the manuscript more clearly, we have revised the titles of different chapters in manuscript. Chapter 4 is entitled "Results", and the secondary titles of this chapter are "Changes in horizontal hydraulic diffusivity" and "Changes in vertical permeability", respectively.

[Figure]

Figure R6. The flow chart.

2. Site description: There should have been a site description instead of what is now mentioned as the geological setting of the area. In the section 5.2 line 390-391 there is also new site descriptive information which needs to be in the site description.

**Response:** Chapter 2 is split into three separated secondary section which are entitled "Regional Geological Setting" "Information of Monitoring Well" and "Data Collection", respectively. The section 5.2 line 390-391 of previous manuscript is moved

3. Chapter 2, consider splitting these in a 'Site description' and a separate 'Methodology and data sources' section. Where in the second section the beforementioned methodology is included.

**Response:** We have split Chapter 2 into three different secondary sections, including Regional Geological Setting, Information of Monitoring Well, and Data Collection. For the methodology, we have added the flow chart of methods in the discussion section. We believe it is the most appropriate way to make reader understand the different methods corresponding to the mechanisms clearly.

4. One could decide to combine the result & discussion (chapter 3-5).

**Response:** Disagree. The content of Chapter 3 describes the observed phenomena of water level and temperature coseismic changes, which is not the model calculation results. It is not appropriate to be merged into the section of the result & discussion. Thus, this section is still a separated chapter.

In the revised manuscript, we revised chapter titles and added secondary titles. The content of Chapter 4 is the results of tidal analysis which estimate the change of vertical and horizontal permeability caused by earthquakes. In Chapter 5, we reveal the coseismic mechanisms of water level and temperature from the static and dynamic strain, and the end-member mixing model. The new title of Chapter 3, 4 and 5 are "Observation of Coseismic Water-level and Temperature Changes", "Results", and "Discussion", respectively. The secondary titles of Chapter 4 are "Changes in horizontal hydraulic diffusivity" and "Changes in vertical permeability". The secondary titles of Chapter 5 are "Mechanisms of Water-level Coseismic Responses", and "Possible Mechanism of Water Temperature Coseismic Change", respectively.

5.Lack of analysis to the magnitude of the proposed mechanisms. There have been issued several mechanisms influencing the hydrological responses, however these only have been appointed to be present. The mechanisms have not been analysed on their

magnitude, which may make a difference in the way the conclusions are drawn, it will probably change the emphasise on a certain mechanism. Though this may be out of scope for this research.

**Response:** Disagree. The content of Chapter 3 describes the observed phenomena of water level and temperature coseismic changes, which is not the model calculation results. It is not appropriate to be merged into the section of the result & discussion. Thus, this section is still a separated chapter.

In the revised manuscript, we revised chapter titles and added secondary titles. The content of Chapter 4 is the results of tidal analysis which estimate the change of vertical and horizontal permeability caused by earthquakes. In Chapter 5, we reveal the coseismic mechanisms of water level and temperature from the static and dynamic strain, and the end-member mixing model. The new title of Chapter 3, 4 and 5 are "Observation of Coseismic Water-level and Temperature Changes", "Results", and "Discussion", respectively. The secondary titles of Chapter 4 are "Changes in horizontal hydraulic diffusivity" and "Changes in vertical permeability". The secondary titles of Chapter 5 are "Mechanisms of Water-level Coseismic Responses", and "Possible Mechanism of Water Temperature Coseismic Change", respectively.

6.The text is not always concise in the use of grammar and spelling, the conclusion for example first starts in the past tense and then switches to the present simple.

**Response:** We have corrected the wrong tense of manuscript, especially in the abstract and conclusion section, and revised the sentences with the present tense. The revised part is highlight in yellow.

In line comments

1.98, where is this 4 m coming from? Please provide elevation level of the well to check based on values in line 81 whether 4m is correctly calculated.

**Response:** The elevation level of Dazhai well is about 1471m. the groundwater head of the shallow sandstone aquifer is about 1475m. Without the discharge port, the hydraulic head would be 4 meters above the surface. We have added the information of

elevation level of the well in Line 82~83 of revised manuscript.

2.Caption Figure 3 & 4, I suggest to change 'times' to 'events'

**Response:** Corrected.

3.Line 136 typo: Mk -> mK

**Response:** Corrected.

4.196 there is for example mentioned something about a 95% confidence interval, however it is not clear to me which values are within this 95% interval. Do you mean that the errors are within a 95% interval or the changes in phase shift and tidal factor? This could either be a typo or an actual research failure.I would recommend providing this statistical test as a supplement.

**Response:** According to the calculation results from Baytap-G, the phase shift and tidal factor inferred from water level have calculation errors. The error bars represent the 95% confidence interval which indicates that there is 95% probability of occurrence for calculation results (phase shift and tidal factor) within the range of confidence interval.

5.199 Incomplete data representation, only for EQ4, 6, 7, 11 the changes are discussed, what are the changes for the other earthquakes?

**Response:** We have added the calculation results in Line 228~229 of revised manuscript. The rate of change in tidal factors changes following EQ 1, 8 and 12. The changes are 50%, 12.5%, and 25% following EQ1, EQ8, and EQ12, respectively.

6.206 & 207, Typo? => there is stated that EQ11 and EQ12 have a phase shift smaller than -10 degree, however according to the table this value is larger than -10 degree.

**Response:** It is a typo. The value of phase shift change in Table 2 means the difference between the phase shift after and before earthquakes. The phase shift following EQ 11 and EQ12 are > -10°. We have revised it in Line 236~239 of revised manuscript.

7.210, typo in maybe it should be 'may be'.

**Response:** Corrected.

8.Figure 5: What do the red and blue dotted lines indicate? Please add this to the legend of the figure.

**Response:** *H* is the amplitude of the fluctuating pressure head in the elastic aquifer responding to the tidal stress. Due to the influence of seismic activities, the value of *H* is change before and after earthquake. The blue and red lines is the theoretical model (equation 2 and 3) with different *H*. We have added the legend of Figure R7 in the revised manuscript.

[Figure]

Figure R7. Amplitude versus phase shift at the frequency of the M2 wave. Dots before earthquake (blue dots) and after earthquake (red dots) fit two curves that represent the theoretical model (equation 2 and 3) with different *H*, respectively.

9.241, 321, 444 preexisting or pre-existing, please be constant in its spelling

**Response:** We have revised these spelling problems with a uniform format.

10.244 word-choice, dilate -> dilation

**Response:** We have revised it.

11.245-247 & table 2, Here indicated that for the next analysis the values in table two will be used, however EQ1 and EQ8 are indicated not to be used but are still in the table. I suggest discarding them from the table

**Response:** The earthquakes in Table 2 are the result of earthquakes classification in Table 1, because we consider earthquakes that occurred within three months of each other as a single earthquake to reduce post-seismic effects on the aquifer properties. If the EQ1 and EQ8 are discard from Table 2, the classification results are inaccurate. In addition, although the large distant earthquake EQ1 and EQ8 with the lower value of static strain are not analyzed in the static strain section, both of them are used to analyze the mechanisms of dynamic strain. Thus, EQ1 and EQ8 in Table 2 cannot be discarded in Table 2.

12.385-389 Consider moving this part to the methodology section

**Response:** Disagree, we have responded it in our previous response.

13.390-392 Info could be added the Site description, this is new information about the area

**Response:** We have moved the climate information into section 2.1 Regional Geological Setting of revised manuscript.

14..Conclusion -> Inconsistent in the used tense, either use past tense or present tense

**Response:** We have corrected the wrong tense of the conclusion section, and revised the sentences with the present tense.

15.445 add "the" before hydrogeological setting

**Response:** We have added it.

16.445 either it is: from 'the deeper' aquifer or from deep 'aquifers'

**Response:** It should be "from the deeper aquifer". We have revised it.

**Reference:**

Allegre, V., Brodsky, E. E., Xue, L., Nale, S. M., Parker, B. L., and Cherry, J. A.: Using earth-tide induced water pressure changes to measure in situ permeability: A comparison with long-term pumping tests, Water Resources Research, 52, 3113-3126, https://doi.org/10.1002/2015WR017346, 2016.

Xue, L., Li, H. B., Brodsky, E. E., Xu, Z. Q., Kano, Y., Wang, H., Mori, J. J., Si, J. L., Pei, J. L., Zhang, W., Yang, G., Sun, Z. M., and Huang, Y.: Continuous Permeability Measurements Record Healing Inside the Wenchuan Earthquake Fault Zone, Science, 340, 1555-1559, https://doi.org/10.1126/science.1237237, 2013.

Zhang, S., Shi, Z., and Wang, G.: Comparison of aquifer parameters inferred from water level changes induced by slug test, earth tide and earthquake - A case study in the three Gorges area, Journal of Hydrology, 579, https://doi.org/10.10.1016/j.jhydrol.2019.124169, 2019.